# Velocity Variability and Performance in Backstroke in Elite and Good-Level Swimmers

**DOI:** 10.3390/ijerph19116744

**Published:** 2022-05-31

**Authors:** Aléxia Fernandes, Márcio Goethel, Daniel A. Marinho, Bruno Mezêncio, João Paulo Vilas-Boas, Ricardo Jorge Fernandes

**Affiliations:** 1Centre of Research, Education, Innovation and Intervention in Sport and Porto Biomechanics Laboratory, Faculty of Sport, University of Porto, 4200-450 Porto, Portugal; alexiaafernandes@gmail.com (A.F.); gbiomech@fade.up.pt (M.G.); jpvb@fade.up.pt (J.P.V.-B.); 2Department of Sports Sciences, University of Beira Interior, 6201-001 Covilhã, Portugal; dmarinho@ubi.pt; 3Research Centre in Sports, Health and Human Development (CIDESD), 6201-001 Covilhã, Portugal; 4Biomechanics Laboratory, School of Physical Education and Sport, University of São Paulo, São Paulo 05508-030, Brazil; brunomezencio@msn.com

**Keywords:** backstroke swimming, intracycle velocity variation, stability, complexity

## Abstract

Backstroke swimming, a cyclic and continuous movement, displays a repeating structure due to the repeated action of the limb, presenting similar (but not identical) cycles. Some variability is generated by instabilities, but this may play a functional role in the human performance, allowing individual adaptations to constraints. The current study examined the role of velocity variability in backstroke performance, hypothesizing that this variable is associated with swimmers’ performance. Sixteen elite and fifteen good-level swimmers were video recorded in the sagittal plane when performing 25 m backstroke at maximal intensity in order to determine hip velocity and mean velocity, stroke rate, stroke length and indexes of coordination/synchronization. Lyapunov maximal exponent and sample entropy were also calculated for successive cycles. The elite swimmers’ performances were more unstable (0.1742 ± 0.1131 versus 0.0831 ± 0.0042, *p* < 0.001) and complex (0.9222 ± 0.4559 versus 0.3821 ± 0.3096, *p* < 0.001) than their good-level counterparts, but intracycle velocity variation did not differ (11.98 ± 3.47 versus 12.03 ± 3.16%, *p* > 0.05). Direct relationships were observed between mean velocity and stability (*r* = 0.40, *p* = 0.03), as well as with complexity (*r* = 0.53, *p* = 0.002), with intracycle velocity variation and complexity also being related (*r* = 0.38, *p* = 0.04). Backstroke performance is associated with velocity variability, with elite swimmers being able to control it through several adaptations, overcoming the high drag and inertia.

## 1. Introduction

Swimming is a cyclic sport that displays a repeating structure due to the continuous action of the limbs. However, swimming also presents biomechanical variability in each one of the conventional propulsive techniques, demonstrating similar but not identical consecutive cycles [1,2]. Many environmental-, individual-, and task-related perturbations modify the ongoing movement dynamics and are considered inherent biological noise. The movement continuity after the perturbations is preserved by adjusting the motion parameters rather than recruiting a new motor pattern [3]. However, since human movement systems are open, non-linear dynamical systems, variability may play a functional role in helping individuals explore the environment [4].

Variability in cyclic movements is a very well-studied topic in sport sciences, especially in swimming, where researchers and coaches focus on intracycle velocity [2,5] and force variations [6,7], upper- and inter-limb coordination variability [8,9], stroke rate and stroke length inter-lap variability [10], electromyography [5,11], and body segments [12,13]. These approaches often use standard deviation, coefficient of variation and mean difference statistical measures to quantify variation, but it is the structure of the variability (not only its magnitude) that is important to better understanding good versus bad performances. The ways of quantifying intracycle velocity variation may seem to be insufficient, since the variation structure gives additional information about the movement variability and provides a more complete performance characterization, requiring the analysis of both the amount and structure of variability [1,4].

The literature presents conflicting results about intra-cyclic velocity variation (commonly assessed using the coefficient of variation), showing both higher and lower values in elite swimmers compared to their lower-level counterparts. However, a lower intra-cyclic velocity variation should produce a more efficient swim for the same drag condition [2,13]. These conflicting data might be explained by individual differences in drag profile and might not be directly related to the velocity variation itself. Studies focusing specifically on backstroke swimming are scarce, and even fewer compare the velocity variation between swimming techniques in age-group competitive swimmers [14,15,16]. More recently, motor control methodologies have been used to analyze the swimming motion variation through stability and complexity calculations, providing new insights into time series variability [16].

Backstroke is an alternated swimming technique characterized by a continuous propulsion or shorter non-propulsive lags [15,17]. Of the four conventional swimming techniques, backstroke is the only one that is performed in the dorsal position, with the shoulder joints’ anatomical configuration interfering with the movement amplitude, leading to lower swimming velocities [18]. Swimmers’ velocity is key to excelling in swimming events, but it is affected by the capacity to propel the body with minimal velocity variation and energy losses [13]. Therefore, the efficiency of the segmental actions should be maximized (to attain the highest stable velocity), but intra-cycle flexible adaptations are still necessary. We aimed to assess the role of backstroke velocity variability in swimmers of both elite and good levels, hypothesizing that variability-related variables are associated with swimmers’ sprint performance.

## 2. Materials and Methods

Sixteen elite (twelve females, with qualifying standards high enough to participate in the World and European Junior Championships’2021, i.e., training ≥ nine ~6000 m sessions/week totaling ~22.5 h) and fifteen good-level swimmers (seven females, with regional and national standards, training ≥ six ~4500 m sessions/week totaling ~15 h) participated in the current study. Their main physical and performance related characteristics (best backstroke event FINA points) are summarized in Table 1.

In a 1.90 m, deep 25 m long indoor swimming pool with a 27 °C water temperature, and after a standardized warm-up [2], swimmers were video recorded while performing 25 m backstroke at maximal intensity. An underwater and an aerial camera (Go Pro 6, San Mateo, CA, USA), operating at 120 Hz sampling frequency, with 1920 × 1080 pixels resolution in wide mode, were fixed to a camera set-up in the sagittal plane and pushed alongside the pool [2]. Ten markers rows were placed on the pool floor (with a 2.5 m distance in between each one), allowing subsequent calibration and coordinates transformation (Figure 1) [2]. 

Data processing was performed in Matlab (MATLAB R2019b, The MathWorks Inc., Natick, MA, USA), and intrinsic camera parameters had been previously calculated to correct distortions. The initial extrinsic parameters were obtained using a six marker rectangular rigid calibration body (2.0 × 1.0 m), with markers placed horizontally at 0, 1.0 and 2.0 m and vertically at 0 and 1.0 m [2,19]. The six markers with known coordinates were digitized, and the DLT method was applied for reconstruction [19], with a final reconstruction error < 1.43 mm. The hip coordinate was digitized and considered as a reference for swimmers’ position assessment [20]. A 0.98 intraclass correlation coefficient (fixed-effects, 2-way ANOVA model [21]) demonstrated high inter- and intra-evaluator reliability for this process. To obtain the hip velocity, the hip horizontal coordinate was filtered using a low-pass fourth Butterworth (7 Hz cut-off frequency) and then differentiated [19]. 

Videos were analyzed and trials were divided by each backstroke’s upper limb cycles, removing the first and the last from the examination. One hundred and ninety-six cycles were analyzed, and the corresponding cycle phases were detected using the Blender v2.79b open source software (Amsterdam, Netherlands). Each phase was identified as follows [15,22]: (i) first down sweep, from the hand entry in water with the elbow in maximum extension; (ii) first up sweep, from the start of the elbow flexion, with the hand moving backwards until it is perpendicular to the shoulder; (iii) second down sweep, from the hand below the shoulder to the end of its backward movement; (iv) second up sweep, from the moment the hand is still at the tight to the water exit; and (v) recovery, from the exit to the new in-water hand entry. Phase detection was carried out with careful observation of the orientation of the hands and elbows, and analyzed frame by frame for validation. A 0.94 intraclass correlation coefficient (fixed-effects, 2-way ANOVA model) demonstrated the high reliability of the inter- and intra-evaluators, which were evaluated in another dataset. Attending to the importance of the synchronization of the lower limb actions, the foot position at each upward action end was also identified.

Backstroke cycle duration, instantaneous velocity, mean velocity, absolute and relative minimum and maximum velocities, stroke rate, stroke length and intra-cycle velocity variation were calculated using Matlab (MATLAB R2019b, The MathWorks Inc., Natick, MA, USA). Stroke rate and stroke length were assessed as the inverse of the cycle duration and the ratio between velocity and stroke rate (respectively [2]), with the upper limb phase duration and lower limb actions expressed as a complete cycle duration percentage. Index of coordination was defined as the lag time between the start of the first upper limb propulsion and the end of propulsion of the contralateral upper limb and was calculated by the difference between timings [22,23]. Accordingly, the propulsive phase duration was the sum of the upper limb first up sweep and second down sweep phases, and the non-propulsive phase was obtained by adding the upper limb first down sweep, second up sweep, and recovery phases. The index of synchronization was defined as the coordination shift between cycles and calculated by the ratio between the lower limb action rate and stroke rate [19]. The difference between this ratio and its nearest integer number resulted in a dimensionless index (ranging between ± 0.5), with positive or negative results showing lower limb delays or advances, while results ~0 indicated no phase shift between cycles. 

The intra-cycle mean velocity coefficient of variation was used to determine intra-cycle velocity variation [24,25,26]. Since nonlinear dynamics are sensitive to motor behaviors and may identify significant variations, stability and complexity were assessed to quantify the irregularity and unpredictability of a temporal structure dataset [16]. Velocity stability was calculated through the Largest Lyapunov Exponent using a previously proposed algorithm [27]. If positive, the nonlinear deterministic system is chaotic, and the greater the value is, the more divergent the attractor is [28]. Velocity complexity was determined through the sample entropy algorithm described in the following equation [29], working as displayed in Figure 2.
(1)$$SampEn=-log\left(\frac{\left(\sum A_{i}\right)}{\left(\sum B_{i}\right)}\right)=-log\left(A/B\right),$$

Post-hoc power analysis presented a 0.99 statistical power, a 1.44 large effect size, and a 0.05 overall significance level between groups (G*Power 3.1.9.7, Heinrich-Heine-Universität Düsseldorf, Düsseldorf, Germany). Descriptive analyses were obtained for all variables and data were checked for distribution normality and variance homogeneity with the Kolmogorov–Smirnoff and Levene tests. An independent measures t-test examined the differences between elite and good-level swimmers, with a *p* < 0.05 being accepted. The chi-squared test was used to examine the distribution of the index of synchronization between groups, and linear correlations were performed between mean velocity, intracycle velocity variation, stability, and complexity. 

A forward stepwise linear regression identified mean velocity, complexity, and stability predictors in elite and good-level groups from intracycle velocity variation, stability, complexity, stroke rate, index of coordination, and the relative duration of phases. Variables were added based on *p*-values, with a 0.10 value threshold used to limit the amount of variables included in the final model. Effect sizes were computed using Cohen’s d (small, moderate and large if = 0.2, 0.5 and 0.8).

## 3. Results

Elite swimmers presented a higher mean velocity and SD than their good-level counterparts (1.54 ± 0.11 versus 1.35 ± 0.15 m.s^−1^, *p* < 0.001, d = 1.44 and 0.18 ± 0.05 versus 0.16 ± 0.04 m.s^−1^, *p* = 0.001, d = 0.44; Figure 3). However, the elite swimmers’ performance (compared to the good-level group) was more unstable (0.1742 ± 0.1131 versus 0.0831 ± 0.0042, *p* < 0.001, d = 1.15) and complex (0.9222 ± 0.4559 versus 0.3821 ± 0.3096, *p* < 0.001, d = 1.39), even if the intracycle velocity variation did not differ (11.98 ± 3.47 versus 12.03 ± 3.16%, *p* > 0.05, d = −0.02). The absolute maximum and minimum velocities were higher in elite than in good-level swimmers (1.93 ± 0.20 versus 1.66 ± 0.21 m.s^−1^, *p* < 0.001, d = 1.32 and 1.17 ± 0.16 versus 1.06 ± 0.16 m.s^−1^, *p* < 0.001, d = 0.9), with their relative values being similar (125 ± 9 versus 123 ± 9%, *p* = 0.10, d = 0.22 and 76 ± 9 versus 79 ± 6%, *p* = 0.04, d = −0.39). Elite swimmers, compared with their good-level peers, presented a higher stroke rate (0.82 ± 0.07 versus 0.71 ± 0.09 cycles.s^−1^, *p* < 0.001, d = 1.36) and stroke index (2.89 ± 0.37 versus 2.57 ± 0.41 m^2^.s^−1^. cycle, *p* < 0.001, d = 0.82) for similar stroke length values (1.88 ± 0.16 versus 1.90 ± 0.17 m. cycle^−1^, *p* = 0.43, d = −0.12).

Elite swimmers showed a shorter first down sweep (14 ± 4 versus 18 ± 9%, *p* < 0.001, d = −0.57) and a longer first up sweep and second down sweep (24 ± 4 versus 22 ± 5%, *p* = 0.001, d = 0.44 and 19 ± 4 versus 18 ± 4%, *p* = 0.02, d = 0.25) than the good-level group, with the second up sweep and recovery phases presenting no differences (12 ± 26 versus 13 ± 10%, *p* = 0.89, d = −0.05 and 30 ± 26 versus 29 ± 6%, *p* = 0.72, d = 0.05; Figure 4). Both groups presented a backstroke catch-up coordination, but elite swimmers displayed lower upper limb time lag (−8.41 ± 5 versus −11.23 ± 6%, *p* < 0.001, d = 0.51). Most of the swimmers presented a null index of synchronization, without differences between groups (*n* = 13 and 14 in elite good-level groups; chi-squared = 0.876, *p* = 0.35; index of synchronization = 0.23 ± 0.03 for the non-synchronized swimmers).

Regarding the performance association with the studied variables when considering the global sample (Figure 5), relevant relationships were observed between the mean velocity, stability (*r* = 0.40, *p* = 0.03), and complexity (*r* = 0.53, *p* = 0.002), with intracycle velocity variation and complexity also being related (*r* = 0.38, *p* = 0.04). However, when considering each swimming level group, a relevant association between mean velocity, stability (*r* = −0.11, *p* = 0.25 and *r* = −0.09, *p* = 0.42) and complexity (*r* = 0.15, *p* = 0.11 and *r* = −0.05, *p* = 0.65) was not observed, even if elite intracycle velocity variation was related to complexity (*r* = 0.47, *p* < 0.001). 

For elite swimmers, stroke rate (*p* < 0.001), intracycle velocity variation (*p* = 0.003), first (*p* < 0.001) and the second down sweep phases (*p* = 0.005) were good mean velocity predictors (*r* = 0.736, R^2^ = 0.541, F = 17.110, *p* < 0.001; Table 2). Intracycle velocity variation (*p* < 0.001) and relative maximum velocity (*p* = 0.04) were good predictors of complexity (*r* = 0.561, R^2^ = 0.314, F = 13.742, *p* < 0.001), but no variables estimated stability. For good-level swimmers, the model was validated (*r* = 0.728, R^2^ = 0.529, F = 50.642, *p* < 0.001), with stroke rate (*p* < 0.001) and second up sweep (*p* = 0.001) being good mean velocity predictors. For complexity (*r* = 0.469, R^2^ = 0.220, F = 8.344, *p* < 0.001), the first (*p* < 0.001) and second down sweep (*p* = 0.006) and relative durations of recovery (*p* = 0.017) were good predictors, and index of coordination (*p* < 0.001) was used to estimate stability (*r* = 0.274, R^2^ = 0.075, F = 7.396, *p* = 0.008).

## 4. Discussion

Swimming performance depends on several individual-, environmental-, and task-related factors, thus making the motion complex, irregular, and unpredictable. Swimmers’ adaptation to those constraints is now considered as a superior ability rather than inherent biological noise [8,9,30]. Stability and complexity are variables that are sensitive to motor behaviors that can quantify the irregularity and unpredictability of temporal signals, providing deeper information about how swimmers manage their sprint velocity [16]. The current study evidenced that backstroke performance is associated with velocity variability and that elite swimmers’ performance was more unstable and complex than that of their good-level counterparts, despite the similar intracycle velocity variation values observed. 

As expected, the elite swimmers’ mean velocity was higher compared to that of their good-level swimmers, since they probably produced higher active drag, which could impose a higher energy expenditure [31]. Considering that intracycle velocity variation results from the interaction between propulsive and drag forces, its lower values imply greater mechanical efficiency for the same drag condition [13]. As such, it is possible to assume that intracycle velocity variation would be lower in more efficient swimmers if the drag characteristics were similar. In this way, similar intracycle velocity variation values may suggest elite swimmers’ higher ability to control these variations, implying several adaptations to overcome the high drag and inertia. This is supported by the current intracycle velocity variation and the complexity association.

Several factors could describe the way higher-ability swimmers manage constraints, which could be related to the individual’s technique and coordination. In the current study, elite swimmers (compared to their good-level peers) evidenced longer duration propulsive phases, which accounted for their higher propulsive force production [17,22]. Additionally, at higher velocities, active drag is higher [31], which reduces the glide time, as observed in the elite group and in accordance with the literature [32]. Thus, the second up sweep relative duration (also known as “clearing phase”) was not different among groups, but, apart from the elite’s higher dispersion, presented a tendency to be shorter than the one presented by the good-level swimmers, which likely improved the simultaneity of the earlier hand water exit during the contralateral first up sweep [23,33]. These alternated actions, together with a profiled body position, likely improved continuity and resulted in a better swimming mechanical efficiency (as evidenced by the observed higher stroke index). 

Our elite swimmers move their upper limbs out of the water more rapidly and have a higher stroke rate, as observed in world-class backstrokers [23]. Moreover, a better body roll (allowing the hand to glide longer) and an extended second up sweep when finishing the cycle account for higher stroke lengths [13,34]. We observed that the elite swimmers upper limb coordination was more continuous, even if the catch up is the exclusive coordination mode for backstroke (due to the limited shoulder flexibility and the specific body roll [18]). Thus, swimmers should minimize their upper limb cycle lag time to compensate for the speed loss during the second up sweep [23,34]. Possible reductions in the shoulder entry angle from the suggested optimum may increase resistive drag due to the lateral deviations, but this can be reduced through improvements in technique [34]. In addition, the index of synchronization demonstrated that both groups maintained a constant pattern in their lower limb action at the same relative time in each backstroke cycle [19].

The current study showed that different levels of swimming performances can be explained by different predictors. Elite swimmers mean velocity was well estimated by stroke rate, first down sweep, intracycle velocity variation, and second down sweep. Good-level swimmers mean velocity was also predicted by stroke rate, since it was directly influenced by swimming at high velocities. In this level group, the relative durations of first and second down sweeps also contributed to mean velocity, allowing swimmers to glide and prolong the time during which propulsive force could be applied. Despite the intracycle velocity variation values being similar between groups, there was observed a small tendency to better predict elite performance, possibly due to the velocity dispersion. Lasty, the second up sweep relative duration also estimated good-level swimmers’ mean velocity, accounting for the maintenance of lower velocities due to natural speed loss [17].

Elite swimmers’ complexity was positively estimated by intracycle velocity variation, as this is a variability indicator. In fact, sprint swimming is a highly constrained task, with many degrees of freedom needing to be controlled; therefore, a variety of movement patterns must be adopted to optimize performance [12]. Complexity increased with lower relative maximum velocity, possibly because the main propulsive phases were better represented by a stable velocity in the elite swimmers than in their good-level counterparts. For example, in Figure 3 it is possible to observe that good-level mean velocity presents two main peak velocities, but the elite group did not display that same velocity increment, which may in turn contribute to the intracycle velocity irregularity. However, this is only speculative and further investigation is required, as a higher cycle amount and more swimmers would be necessary to ascertain this hypothesis.

Good-level swimmers’ complexity values, estimated by first and second down sweep and recovery relative durations, suggest that shorter gliding, force production, and recovery phases possibly disarrange the simultaneity between the upper limbs, making the global backstroke swimming technique more unpredictable. Stability reflects the individual ability to recover from small perturbations, with the absence of predictors for the elite swimmers’ group emphasizing the highly constrained performance. The index of coordination was a good estimator for the good-level swimmers’ group, possibly due to the lower velocity, presenting fewer upper peak velocities. Indeed, good-level swimmers were able to recover from perturbations, but the task was not so demanding as in elite sprinting due to the lower mean velocity (and theoretically drag). 

The main findings of the current research benefit swimmers and coaches by giving new insights into the velocity variability role in different level backstrokers. Considering that both groups presented similar intracycle velocity variation, elite swimmers likely adopted strategies to manage velocity variations along the highly constrained environment (aquatic) and task (sprint). In fact, it is becoming an accepted fact that movement variability is an emergent behavior under several constraints regarding inter-limb coordination [35,36] and spatiotemporal kinematics [12,16,37]. Despite the novelty of this study, there are other ways to assess swimming velocity, some of which are more accurate and complete, but also more time-consuming (e.g., motion capture systems [20]). In addition, the coefficient of variation used seems to limit the velocity variation quantification because it does not consider drag force. Future research should deeply analyze variability indicators in the other competitive swimming techniques, aiming to explore their functional role in performance and ascertain the variability behavior in a wider range of swimming levels and ages.

## 5. Conclusions

This study determined other variables to quantify velocity variability in two backstroke swimmers’ levels. Stability and complexity were associated with backstroke performance, and elite swimming was found to be more unstable and complex. Since intracycle velocity variation was similar between group levels, the best swimmers evidenced strategies to control their variations and maintain high performances. Coaches and swimmers should acknowledge that these strategies are possibly related to better and smoother backstroke technique, which implies lower drag forces while swimming at higher velocities. In this way, practicing a “smoother” swimming technique should be key in the training process. 

## Figures and Tables

**Figure 1 ijerph-19-06744-f001:**
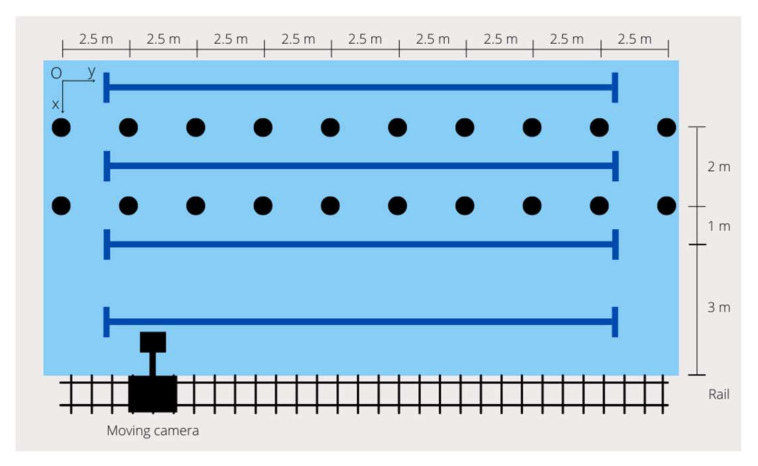
Protocol set up used to record swimmers’ performance (the markers were placed on the pool floor).

**Figure 2 ijerph-19-06744-f002:**
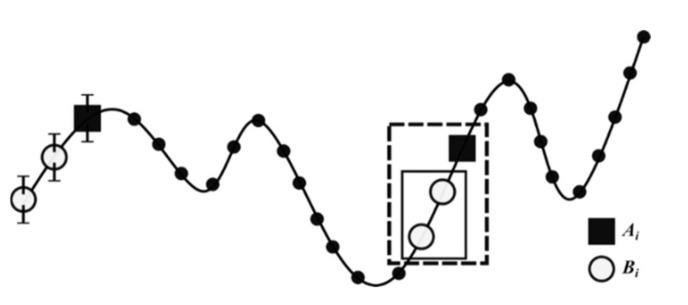
Schematic demonstration of entropy estimation using sample entropy, with the time series beginning with the *i*th template. *A_i_* = number of matches of length *m* + 1 with *i*th template and *B_i_* = number of matches of length *m* with *i*th template. The parameter *m* is 2 and the tolerance for accepting matches is *r* = 0.2 times the SD (error bars). The template is matched by the 16 and 17th points (solid box), and the *m* + 1st points also match (dashed box). Therefore, *A* and *B* increase by 1.

**Figure 3 ijerph-19-06744-f003:**
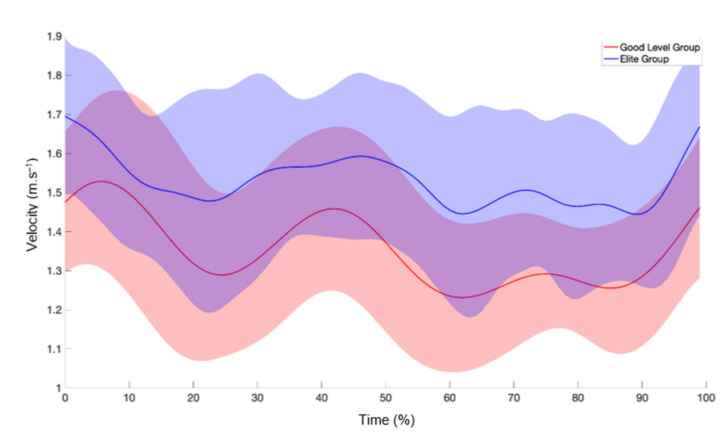
Elite and good-level groups’ mean velocities and SDs (blue and red lines and shades, respectively).

**Figure 4 ijerph-19-06744-f004:**
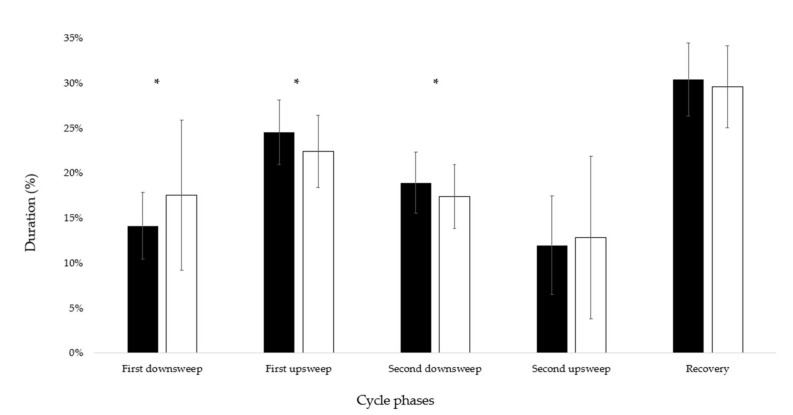
Relative duration of elite and good-level swimmers’ backstroke cycle phases (black and white colors, respectively). * Represents differences between groups.

**Figure 5 ijerph-19-06744-f005:**
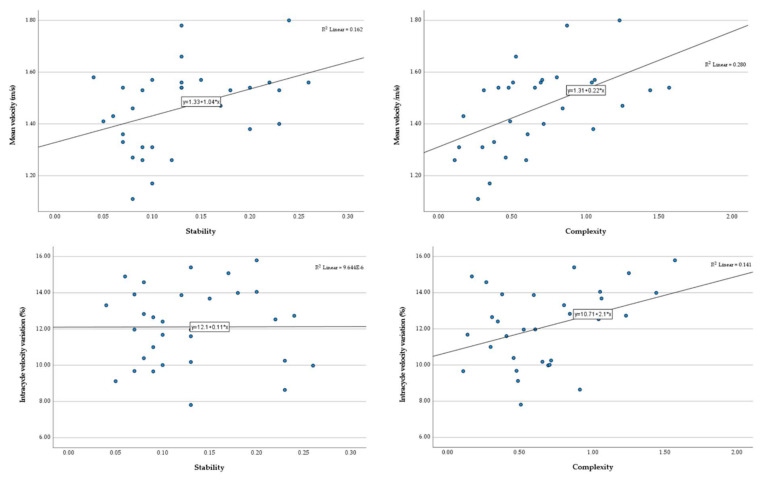
Relationship between mean velocity and stability, mean velocity and complexity, intracycle velocity variation and stability, and intracycle velocity variation and complexity (upper left and right, and lower left and right panels, respectively).

**Table 1 ijerph-19-06744-t001:** Main physical and performance characteristics of elite and good-level swimmers (mean ± SD).

Variables	Elite			Good Level			Pooled Data		
	Female	Male	*p*	Female	Male	*p*	Elite	Good Level	*p*
Age (years)	15.9 ± 1.0	17.0 ± 0	0.05	15.1 ± 1.2	16.3 ± 1.3	0.11	16.2 ± 1.0	15.7 ± 1.3	0.29
Height (cm)	167.8 ± 2.7	177.5 ± 1.9	0.00	162.6 ± 6.9	177.5 ± 4.8	0.00	170.3 ± 5.0	170.5 ± 9.6	0.92
Body mass (kg)	57.8 ± 4.0	65.1 ± 3.8	0.00	54.7 ± 5.7	65.1 ± 10.2	0.03	59.6 ± 5.0	60.3 ± 9.7	0.81
FINA points	713 ± 117	709 ± 155	0.96	360 ± 53	353 ± 65	0.80	712 ± 122	356 ± 58	0.00

**Table 2 ijerph-19-06744-t002:** Step-wise regression coefficients in elite and good-level swimmers used for predicting mean velocity, complexity, and stability.

			Unstandardized Coefficients		Standardized Coefficients	95% Confidence Interval	
			B	Std Error	Beta	Lower Bound	Upper Bound
	Mean velocity	(Constant)	0.245	0.161		−0.077	0.567
Elite		Stroke rate	1.490	0.191	1.002	1.108	1.872
		1st down sweep	0.921	0.240	0.381	1.441	1.402
		Intracycle velocity variation	0.009	0.003	0.305	0.003	0.015
		2nd down sweep	−0.881	0.300	−0.367	−1.480	−0.281

	Complexity	(Constant)	1.854	0.828		0.197	3.511
		Intracycle velocity variation	0.095	0.021	0.791	0.054	0.136
		Relative maximum velocity	−1.654	0.803	−0.353	−3.260	−0.049
Good level	Mean velocity	(Constant)	0.493	0.087		0.321	0.665
		Stroke rate	1.132	0.115	0.720	0.904	1.361
		1st up sweep	0.368	0.105	0.256	0.159	0.578

	Complexity	(Constant)	0.575	0.174		0.229	0.921
		1st down sweep	−1.304	0.323	−0.379	−1.947	−0.662
		2nd down sweep	2.161	0.761	0.282	0.648	3.674
		Recovery	−1.199	0.494	−0.241	−2.180	−0.218

	Stability	(Constant)	0.062	0.008		0.046	0.079
		Index of coordination	−0.179	0.066	−0.274	−0.309	−0.048

## Data Availability

Not applicable.

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
