# Peer review of "Velocity Variability and Performance in Backstroke in Elite and Good-Level Swimmers"

_ijerph, 2022, doi:10.3390/ijerph19116744_

Round 1
Reviewer 1 Report
I like the methods and result section a lot, but I would like you to ask to improve the end of the introduction and discussion section.
In my opinion, you need to clarify why you are the role of velocity variability is that huge compared to the variation in in other parts of the human being. Please add a few sentence or a paragraph at the end of the introduction.
I would also like to address that it would be an upgrade if you add variation in perfomance into the discussion. Are there any connections between you finding and findings in the literature dealing with "variation in performance" in swimming.
Reviewer 2 Report
Comments and Suggestions for Authors:
Abstract. It is recommended to expand the conclusion. The abstract conclusions do not provide answers to the study's objectives. what about Velocity variability and performance in elite and good level backstrokers? Also, be more specific with the conclusions of the study.
Keywords: The keywords should not appear in the title.
Introduction. Is correct.
Why haven't the groups of men and women been compared?
Materials and Methods. Line 76. What are the differences between the elite group and the good level group?
Line 82. The method section must explain the FINA points.
Line 111. What statistical test was used to demonstrate the validity of ICC3, inter and intra evaluators?
Results. The result and tables are correct.
Discussion. Line 221-223; 337, 339- Supplement these statements with the data from the studies cited.
Conclusions. Correct. Practical applications for trainers should be provided.
References. Correct.
Making the indicated modifications, the study is novel and interesting.
Reviewer 3 Report
Spelling, grammar and scientific vocabulary. English need a deep review. Some examples:
- Line 15-17. Rewrite the sentence (difficult to understand)
- Line 18-20. Rewrite the sentence (difficult to understand)
- Line 24. "Swimmers" for "swimming"
- Line 57. Poor English. Rewrite and use scienttific vocabulary
- Line 65. Change "From"
- Line 302. "groups levels" for "group level"
Participants
Detail training time and volume of participants. Nº of Sessions per week is not enough to define training
Tables and figures
- FINA points. Is that average? Best event?
- No point in comparing males and females' FINA points, height and body mass (delete it)
- Figure 1. Add a key
- Figure 3. Improve image quality and alignement
Design
- For a better understanding include a graph clarifying the recording protocol and the pool refrences
- Many variables are not well defined such as complexity or stability. (i.e. the meaning of velocity complexity is not found until line 269 (disucussion)
- Why do you include the steplinear regresion for predicting complexity and stability as well? The only interesting variable for performance is Mean velocity. The other variables should be included as DVs.
- A table comparing good level vs elite swimmers for all variables (SR, SL, Index of coordination, stability......) should be included
- Include limitations of the study
Round 2
Reviewer 3 Report
The study has improved with the suggestions and comments from reviewers. However there are still some issues.
1) The title is confusing. It should be "Velocity variability and performance in backstroke in elite and good level swimmers" or similar. I wouldn't use the word "backstroker". It is difficult to believe that there were 16 elite backtrokers in the study, especially considering the FINA points in that event (713 ± 117 and 709 ± 155). Junior backtrokers at european level (as you indicate) are usually over 800 FINA points. Probably, the swimmers were at European level but not all of them were bacstrokers. Please fix it to avoid confusion.
2) Figure 1 is a picture of a swimmer performing dolphin kick not gliding
3) The graph or figure with the recording protocol (where the cameras were placed, angles, markers....) has nor been included
